# Preoperative Factors Associated with Target Lesion Revascularization following Endovascular Therapy of the Superficial Femoral Artery

**DOI:** 10.3390/jcm11154606

**Published:** 2022-08-08

**Authors:** Mathias Kaspar, Alexander Bott, Aljoscha Rastan, Joern Fredrik Dopheide, Heinz Drexel, Marc Schindewolf

**Affiliations:** 1Division of Angiology, Swiss Cardiovascular Center, Inselspital, Bern University Hospital, University of Bern, 3010 Bern, Switzerland; 2Division of General Internal Medicine, Inselspital, Bern University Hospital, University of Bern, 3010 Bern, Switzerland; 3Department of Internal Medicine, Subdivision of Angiology, Lucerne Cantonal Hospital, 6006 Lucerne, Switzerland; 4Vorarlberg Institute for Vascular Investigation and Treatment (VIVIT), Carinagasse 47, A-6800 Feldkirch, Austria; 5Private University of the Principality of Liechtenstein, 9495 Triesen, Liechtenstein; 6Drexel University College of Medicine, Philadelphia, PA 19129, USA

**Keywords:** PAOD, superficial femoral artery, endovascular treatment, non-invasive predictors of risk factors, CD-TLR

## Abstract

**Objectives:** In patients with symptomatic peripheral arterial occlusive disease (PAOD), endovascular revascularization of the superficial femoral artery (SFA) is the most frequent intervention. A major drawback of endovascular procedures is clinically driven target lesion revascularization (CD-TLR), which may cause recurrence of symptoms, re-hospitalizations, and re-interventions. Outcome studies comparing endovascular modalities are heterogeneous and focus more on intraoperative rather than preoperative aspects. Studies have not examined potential risk factors in patients’ phenotype before an intervention to prevent CD-TLR. **Design:** Monocentric, retrospective cohort study of 781 patients with symptomatic PAOD referred to an endovascular intervention of the SFA between 2000 and 2018. **Methods:** The study aim was to identify risk factors and phenotypes leading to symptomatic PAOD in patients with de novo lesions of the SFA and ≥1 CD-TLR within 12 months post-index procedure. Two groups were differentiated: patients without CD-TLR and with ≥1 CD-TLR. Patient phenotype was compared for cardiovascular (CV) risk factors, age, gender, and renal function. **Results:** 662 patients (84.8%) (age 73.5 ± 11.2 years; 243 women (36.7%)) with no CD-TLR were compared to 119 patients (15.2%) with ≥1 CD-TLR (age 70.9 ± 12.4 years; 55 women (46.2%)). Women, as well as subjects with dyslipidemia, had each a 1.8-time higher odds ratio of receiving multiple interventions within one year than men or subjects without dyslipidemia. Older subjects (per decade) had a lower odds ratio (0.7) for multiple interventions. Subjects with an eGFR (estimated glomerular filtration rate) <30 mL/min had 3.8 times higher and subjects with eGFR ≥30 and <60 mL/min had a 2.4 higher odds ratio of receiving multiple interventions than subjects with eGFR values ≥90 mL/min. **Conclusion:** Our data indicate that younger women, patients with dyslipidemia, or those with renal insufficiency are at risk for recurrent midterm CD-TLR after endovascular therapy of the SFA.

## 1. Introduction

Endovascular revascularization is a mainstay of therapy in patients with peripheral arterial occlusive disease (PAOD) [1]. The superficial femoral artery (SFA) is the most treated vascular segment in patients with symptomatic lower extremity artery disease [2]. Endovascular treatment became widespread among these patients and is recommended by national and international guidelines as the primary revascularization method [1,3,4,5].

Despite the improvement of catheter-guided procedures and endovascular innovations, clinically driven target lesion revascularization (CD-TLR) remains the main adverse outcome event [5,6,7,8].

Regarding the occurrence of CD-TLR, different pathologies can be distinguished at time of onset. Early CD-TLR (within 30 days after primary intervention) is usually caused by thrombosis due to local pathologies (e.g., incomplete revascularization, residual dissection, or endothelial injury causing local hypercoagulability) [9]. In contrast, late CD-TLR is caused by vascular calcification and is closely linked to aging, diabetes, and chronic kidney disease. Its prevalence and pathophysiologic mechanisms in PAOD are poorly understood. Heavily calcified lesions are usually excluded from investigational device trials [10].

Between the two extremes of thrombosis and calcification, the symptomatic CD-TLR might be affected by risk factors. Cardiovascular risk prevention is an important therapy in the progression of atherosclerosis in PAOD [10] and therefore might improve CD-TLR years after an intervention. CD-TLR at 12 months is usually well recorded as the primary endpoint in interventional trials, but not well characterized by patients’ phenotypical risk factors. It may occur in the context of intimal hyperplasia with vascular remodeling leading to vessel wall thickening and consecutive luminal narrowing [11]. It is not clear whether classical cardiovascular risk factors or potentially other pathomechanisms, e.g., hemostasis [12,13], inflammation [14], and metabolic status [15,16], contribute to this mid-term CD-TLR.

The aim of the present study is to examine the impact of classical cardiovascular and possibly other risk factors on the rate of mid-term CD-TLR between 4 weeks and 12 months after primary recanalization of the SFA.

## 2. Methods

### 2.1. Study Design and Patient Cohort

We performed a single-center, retrospective cohort study analyzing all catheter-based primary interventions of the SFA carried out between 2000 and 2018 during hospitalization in the Department of Angiology at the University Hospital Bern, Switzerland.

The database and participant informed consent form were approved by the Swiss Ethics Committee on research involving humans according to the Declaration of Helsinki (ID 2018-00682). Each patient’s written informed consent was obtained prior to inclusion.

The primary outcome was target lesion revascularization between 1 and 12 months. Patients with re-interventions within the first four weeks of the initial intervention were excluded. Patients were stratified into 2 groups according to CD-TLR after the primary revascularization (“no CD-TLR”, “≥1 CD-TLR”). Patients underwent programmed on-site follow-ups 1 day, 2–4 weeks (optional), 3, 6, and 12 months after the primary revascularization. Monitoring comprised clinical examination (Rutherford stages), ankle-brachial or toe-brachial index, oscillography, and sonography. The decision for CD-TLR was clinically driven (Rutherford ≥ 2 or increase in Rutherford > 1) and confirmed by sonographic and interventional reports.

We selected the following health-related data for our data query: age, sex, body mass index (BMI), history of cardio- and cerebrovascular events (both considered stable, free of event > 12 months), arterial hypertension (mean BP > 135/85 mmHg a/o anti-hypertensive treatment), diabetes mellitus (Hba1c > 5.9% a/o antidiabetic treatment), smoking status ((nonsmokers, current smokers, ex-smoker (>6 months)), C-reactive protein (CRP), HbA1c, creatinine, estimated glomerular filtration rate (eGFR), total cholesterol, HDL cholesterol (HDLC), LDL cholesterol (LDL-C), and triglycerides. Dyslipidemia was defined as LDL-C > 1.8 mmol/l. Data were retrieved upon request by the Insel Data Coordination Lab (IDCL) from electronic health records stored in ClinicWinData (E&L medical systems, Germany), PACSIDS7 (Sectra, Sweden), i-pdos (CompuGroup Medical Schweiz AG, Switzerland), ixserv. 4 (ixmid Software Technologie, Köln, Germany) of the University Hospital Bern.

### 2.2. Inclusion Criteria

Adult patients aged >18 years with symptomatic PAOD of the lower extremities (clinical stage Fontaine II–IV or Rutherford 1–6), initially treated in the Division of Angiology, University Hospital Bern, Switzerland, by endovascular therapy of the SFA, completed follow-up after 12 months and had a complete electronic health record according to our data query mentioned above.

### 2.3. Exclusion Criteria

Incomplete patient record and refusal to study participation.

## 3. Statistical Analysis

### 3.1. Primary Analysis

We removed all variables with more than 30% missing values from the dataset for all further analyses and used all variables to impute missing values in all other variables, using predictive mean matching for continuous variables, logistic regression for binary variables, and the Bayesian polytomous regression model for categorical variables with more than two levels (i.e., smoking). We used a logistic regression model to find risk factors that discriminate between the two groups with either only a single intervention or several interventions using the intervention group as outcome and all potential risk factors as explanatory variables. In a backwards selection approach, starting with the full model intervention group ~age + gender + age × gender + site + BMI + year of PTA + diabetes + hypertension + coronary artery disease (CAD) + cerebrovascular disease (CVD) + dyslipidemia + smoking + triglycerides + LDL-C + HDL-C + HbA1c + eGFR. We performed a backwards selection based on the *p*-value from the likelihood method. We used a *p*-value of 0.15 as a criterion to keep variables during the model selection approach. The final model included intervention group ~age + gender + dyslipidemia + eGFR. EGFR values were recalculated from gender, age, and creatinine clearance after multiple imputations according to the formula by Levey et al. [17]. LDL-C was calculated by the Friedewald formula from total cholesterol, HDL-C, and triglycerides. Because this formula is only applicable if triglycerides are below 4.3 mg/dL, non-calculable LDL-C values were imputed.

### 3.2. Sensitivity Analysis

Sensitivity analyses were carried out to show the robustness of the results in terms of time of laboratory measurements, handling of missing values, and criterion for variable selection.

We performed three sensitivity analyses. First, we used the same approach as in the primary analysis, but with laboratory values within a time window of 180 days from the first intervention. Second, we did an available case analysis with the same model as in the primary analysis, i.e., all missing values in the variables included in the model were dropped from the data. Third, we performed a backwards model selection using the Akaike information criterion (AIC) on a complete case data set.

### 3.3. Missing Values

Baseline variables were almost complete with a maximum of 4% of missing values for BMI (Appendix A). Laboratory parameters had mostly less than 20% of missing values with a minimum of nearly 0% for creatinine P and eGFR and up to 60% for CRP (Appendix A)

### 3.4. Software

All analyses were performed using R version 3.5.0 (R Core Team, R Foundation for Statistical Computing, Vienna, Austria), with the package mice for multiple imputations.

## 4. Results

### 4.1. Patient Cohort and Frequency of Interventions

One thousand and nine patients received an intervention of the SFA during the observational period, 1068 of these patients met the inclusion criteria, and 287 refused participation. Six hundred and sixty-two patients had a single intervention and 119 patients had at least one CD-TLR within 365 days (Table 1).

### 4.2. Primary Analysis of Classical and Non-Classical Risk Factors

In the primary analysis, the two groups differed only in regard to gender or age, with more female patients in the re-intervention group (CD-TLR: 46.2% vs. noCD-TLR: 36.7%; *p* = 0.05), but patients in this group were significantly younger (CD-TLR: 70.9 ± 12.4 vs. noCD-TLR: 73.5 ± 11.2; *p* = 0.024) an received more often lipid-lowering therapy ((CD-TLR: 86.6% vs. noCD-TLR: 77%; *p* = 0.021)). The traditional risk factors diabetes, hypertension, and smoking status (including never, ex-, or active smoker) did not differ between the two groups with either a single intervention or more than one intervention. The same applied to the history of cardio- and cerebrovascular events.

Within 30 or 180 days of the corresponding intervention, CRP, lipids, including total cholesterol, LDL-C, HDL-C, and triglycerides, as well as HbA_1c_, did not differ between the two groups (Table 2). These factors were excluded from further analyses.

### 4.3. Risk for CD-TLR

Female subjects and subjects with pharmacologically treated dyslipidemia (statin a/o ezetimibe) had 1.8 times higher odds of receiving multiple interventions within one year than male subjects or subjects without dyslipidemia (Table 3), even though the lipid levels did not show a significant difference among groups in the primary analysis (Table 2). Furthermore, the risk for multiple interventions increased pronouncedly with lower eGFR values. Particularly, subjects with an eGFR below 30 [mL/min] had 3.8 times higher odds of receiving multiple interventions than subjects with eGFR values above 90 [mL/min] (Table 3). The risk was still markedly increased for subjects with eGFR values between 30 and 90, compared to those with eGRF values above 90. Older subjects had a lower odds for multiple interventions (Table 3).

### 4.4. Secondary and Sensitivity Analysis of Risk for Multiple Interventions

Generally, the results of the primary analysis were well supported by sensitivity analyses. In all sensitivity analyses, the same risk factors were selected during model selection for the final model, except for the addition of the interaction between gender and age in the complete case analysis (Table 4). In the first two sensitivity analyses, female gender had a similar increased risk for multiple interventions as in the primary analysis, and older age had the same ‘protective’ effect (Table 4). In the complete case analysis, the interaction term between age and gender was included but showed no significant effect. All the other risk factors, i.e., dyslipidemia and eGFR, have a very similar odds ratio among the different analyses (Table 4).

## 5. Discussion

The aim of this study was to characterize risk factors in patients with symptomatic PAOD and endovascular treatment of the SFA that predispose them to mid-term CD-TLR within 1 year after primary intervention. Non-invasive risk prediction models for CD-TLR in PAOD are not addressed in current guidelines [1,3].

The results from our primary analysis showed a 1.8 times higher odds ratio of receiving multiple interventions within one year for female or dyslipidemic subjects compared to male or nondyslipedemic subjects (Table 3).

The MARIS Registry enrolled prospectively 998 “real world” patients (657 men; mean age 67.4 ± 9.2 years) with symptomatic SFA stenosis. The primary endpoint was the need for CD-TLR at 12 months which was reached by 136 (17.2%) patients. In accordance with our results, multivariate analysis showed female gender (male vs. female, OR 0.5, 95% CI 0.3 to 0.7, *p* < 0.001) as a predictor of CD-TLR at 12 months, but in contrast to our cohort hypercholesterolemia was not found predictive [18]. We believe this to be explained in part by the higher prevalence of dyslipidemia in our cohort (in total: 66.5% vs. 77.5%).

The increased risk of female gender and younger age for multiple interventions goes well in line with the results from a study by Suzuki et al. [19], who performed a retrospective multicentric analysis of patients (*n* = 432) with de novo SFA lesions treated with the S.M.A.R.T. Control^TM^ stent and identified female gender (42% vs. 26%, *p* < 0.01) and younger age (70.7 ± 9.3 years vs. 72.9 ± 9.0 years, *p* < 0.05) as independently predictive for re-stenosis. However, an inclusion of the interaction term between age and gender like in our complete case analysis, with a generally higher risk for multiple interventions in young females, was not performed.

Iida et al. conducted a survey of 585 consecutive patients receiving endovascular therapy for de novo SFA lesions and divided their cohort into three groups: no, early, and late restenosis [20]. The primary and secondary patency rates up to 6 years indicated that restenosis predominantly occurs within one year. The authors found no significant correlation for age, but for female gender (female 28% (210), 25% (141), 38% (54), 36% (15) (*p* = 0.0071), and the presence of diabetes mellitus (*p* = 0.0428) being more prevalent for early re-stenosis. This supports our result of female gender being predictive for CD-TLR, but not for younger age or conventional risk factors. Apart from dyslipidemia (*p* = 0.021), classical cardiovascular risk factors in our cohort—including diabetes mellitus, smoking, and LDL-C levels—do not differ between the two groups with either none or at least one CD-TLR. These factors seem not to be important for CD-TLR in our collective but are of course highly relevant for the overall cardiovascular risk stratification and outcome and should therefore be treated according to current guidelines [1,3,21].

The risk for multiple interventions increased substantially with lower eGFR values in our cohort. Compared to those with eGRF values above 90 mL/min, subjects with an eGFR below 30 had 3.8 times higher odds of receiving multiple interventions than subjects with eGFR values above 90 mL/min (Table 3). The risk was still markedly increased for subjects with eGFR values between 30 and 90 mL/min. A retrospective analysis from the Zilver PTX Japan Post-Market Surveillance Study evaluated freedom from CD-TLR and patency in patients with or without chronic renal failure at 2 years [22]. The two groups were similar in terms of lesion length and frequency of instent reintervention with similar patency and CD-TLR. This led to the conclusion, that using DES in femoropopliteal artery lesions in chronic renal failure (CRF) patients is safe and effective. In contrast to our study, CRF was simply defined as eGFR < 60 mL/min without further description of absolute values or degree of severity. Thus, according to our results, a reduced eGFR might be associated with a higher odds for multiple interventions, independent of the interventional approach.

In the ZILVER-PTX trial [23] by Zeller et al., renal function was not included in their model to evaluate the association between potential risk factors and loss of patency, even though CRF is a risk factor for CD-TLR and decreased limb salvage.

Patel et al. performed a retrospective cohort study of PAOD patients with infrainguinal vascular interventions divided into two groups for comparative analysis: severe CKD (class 4 and 5; eGFR < 30 mL/min/1.73 m^2^) vs. moderate CKD (eGFR ≥ 30 mL/min/1.73 m^2^) [24]. Their multivariable logistic regression modeling did not show any association between increased late re-interventions at 1 year, neither for severe nor for moderate CKD, which could be explained by a different distribution among baseline characteristics. Prevalence of diabetes was lower in our cohort, and we did not record insulin therapy or hemodialysis, whereas age, gender, and hypercholesterolemia showed similar numbers. It is therefore necessary to address renal insufficiency as a possible risk factor for mid-term CD-TLR in a more precise manner.

The advantage of our retrospective “all inclusive” analysis allows the outcome to be examined for the influence of many clinical variables under “real-life” conditions of clinical practice. However, a retrospective approach to analysis could potentially lead to observational and selection biases besides other limitations.

The effect of referral bias patterns that exist at individual institutions compared to a multicentric approach leaves a smaller variation in different treatment approaches due to a limited amount of interventionalists and creates a limited generalizability to a larger target population, especially for international comparisons.

The inclusion of our patients after an intervention creates an inherent selection bias as patients with severely advanced disease and poor overall prognosis may have been excluded from intervention. Confounding changes in technical aspects during the relatively long inclusion period as well as those patients not being followed in our study center introduce another selection bias.

Due to our focus, the study does not address morphological or anatomical aspects of the culprit lesion. It is unknown in which way these features might confound study results, e.g., vessel diameter in female subjects or degree of calcification in renal insufficiency.

Further, numbers of missing values were very high in some laboratory variables and only specific medication was captured.

Keeping these limitations in mind, we were able to show that between 1 and 12 months after intervention of the SFA subjects of younger age, of female gender, and with dyslipidemia have a higher odds ratio of receiving multiple interventions. Furthermore, the risk for multiple interventions increased substantially with lower eGFR.

In conclusion, our findings show a different pattern of predictive factors for mid-term CD-TLR. As a consequence of everyday clinical practice, patients need to be thoroughly informed about the potential risk of frequent CD-TLR based on their individual risk patterns. Furthermore, future prospective interventional studies need a more consistent and detailed query of patient-specific risk factors.

## Figures and Tables

**Table 1 jcm-11-04606-t001:** Number and percentage of interventions.

Interventions	*n* = 781	%
1	662	84.7
2	93	11.9
3	20	2.6
>3	6	0.8

**Table 2 jcm-11-04606-t002:** Baseline characteristics including risk factors, and laboratory values up to 30 or 180 days after the primary intervention for both groups (no CD-TLR vs. CD-TLR).

Baseline	*n*	No CD-TLR	*n*	CD-TLR	Mann–Whitney Statistic, Mean Difference or Risk Difference (95% CI)	*p*-Value
Gender (female)	662	243 (36.7)	119	55 (46.2)	−9.5 (−19.2 to 0.2)	0.05
Age (at first intervention)	662	73.5 ± 11.2	119	70.9 ± 12.4	2.55 (0.330 to 4.77)	**0.024**
BMI [kg/m^2^]	637	25.7 ± 4.71	119	25.8 ± 4.46	−0.104 (−1.02 to 0.811)	0.82
Target limb	658		119			0.43
left		314 (47.7)		62 (52.1)	−4.4 (−14.1 to 5.4)	
right		344 (52.3)		57 (47.9)	4.4 (−5.4 to 14.1)	
Stent (index procedure)	662	403 (60.9)	119	98 (82.4)	2.99 (1.850 to 4.844)	**<0.0001**
Diabetes	655	224 (34.2)	119	48 (40.3)	−6.1 (−15.7 to 3.4)	0.21
Hypertension	658	567 (86.2)	119	107 (89.9)	−3.7 (−9.8 to 2.3)	0.31
Smoking status	641		119			0.80
never		236 (36.8)		44 (37.0)	−0.2 (−9.6 to 9.3)	
ex		167 (26.1)		34 (28.6)	−2.5 (−11.3 to 6.3)	
active		238 (37.1)		41 (34.5)	2.7 (−6.6 to 12.0)	
CAD	655	249 (38.0)	119	51 (42.9)	−4.8 (−14.5 to 4.8)	0.36
CVD	657	71 (10.8)	119	14 (11.8)	−1.0 (−7.2 to 5.3)	0.75
Dyslipidemia/statin/ezetimibe	652	502 (77.0)	119	103 (86.6)	−9.6 (−16.5 to −2.6)	**0.021**
**Within 30 days**						
Creatinine P [μmol/L]	658	83.5 [68.0, 110]	118	84.5 [69.0, 117]	0.472 (0.417 to 0.529)	0.34
eGFR (calculated) [mL/min]	658	68.5 ± 24.8	118	66.0 ± 27.1	2.51 (−2.43 to 7.45)	0.32
CRP [mg/L]	257	24.0 [7.00, 59.0]	44	10.5 [4.00, 54.5]	0.567 (0.475 to 0.654)	0.15
Triglyceride [mmol/L]	540	1.48 [1.08, 2.14]	101	1.63 [1.13, 2.49]	0.459 (0.399 to 0.520)	0.19
Cholesterol [mmol/L]	541	4.42 ± 1.43	101	4.40 ± 1.23	0.014 (−0.284 to 0.312)	0.93
LDL-C [mmol/L]	541	2.28 ± 1.17	100	2.23 ± 1.03	0.046 (−0.200 to 0.292)	0.71
HDL-C [mmol/L]	542	1.28 ± 0.426	100	1.28 ± 0.469	0.000 (−0.092 to 0.093)	1.00
HbA1c [%]	525	6.00 [5.70, 6.70]	100	6.00 [5.60, 6.78]	0.499 (0.438 to 0.560)	0.98
**Within 180 days**						
Creatinine P [μmol/L]	657	83.0 [68.0, 110]	119	85.0 [69.0, 115]	0.472 (0.417 to 0.529)	0.34
eGFR	657	68.5 ± 24.8	119	66.0 ± 27.0	2.53 (−2.39 to 7.45)	0.31
CRP [mg/L]	322	18.0 [5.00, 54.0]	68	16.0 [5.00, 46.5]	0.532 (0.457 to 0.606)	0.40
Triglyceride [mmol/L]	557	1.48 [1.08, 2.14]	110	1.61 [1.14, 2.46]	0.463 (0.405 to 0.522)	0.22
Cholesterol [mmol/L]	559	4.42 ± 1.43	110	4.36 ± 1.24	0.057 (-0.230 to 0.345)	0.70
LDL-C [mmol/L]	558	2.30 ± 1.18	110	2.24 ± 1.00	0.069 (-0.168 to 0.305)	0.57
HDL-C [mmol/L]	558	1.28 ± 0.426	110	1.27 ± 0.459	0.010 (-0.078 to 0.099)	0.82
HbA1c [%]	545	6.10 [5.70, 6.70]	106	6.15 [5.70, 7.10]	0.487 (0.427 to 0.546)	0.66

Reported are n (%) for categorical variables, and mean ± sd, or median [lower quartile, upper quartile] for continuous variables, with the corresponding risk difference (in %), mean difference, or Mann–Whitney statistic. Categorical variables are compared with the Fisher exact test, continuous variables with the t-test, or the Mann–Whitney U test. The Mann–Whitney statistic reports the probability of a random value taken from one group to be larger than a value in the other group, and ranges from 0 to 1. If there is no difference between the groups, the Mann–Whitney statistic is 0.5. CAD = coronary artery disease; CVD = cerebro-vascular disease; eGRF = estimated glomerular filtration rate (calculated); CRP = C-reactive protein; LDL-C = low-density lipoprotein cholesterol; HDL = high-density lipoprotein cholesterol; BMI = body mass index.

**Table 3 jcm-11-04606-t003:** Odds ratio to receive CD-TLR for all risk factors in the final model.

Risk Factor		*p*-Value
Female gender	1.75 (1.15 to 2.66)	**0.009**
Age (per decade)	0.67 (0.54 to 0.83)	**<0.001**
Dyslipidemia	1.83 (1.03 to 3.25)	**0.039**
eGFR (=60 and <90 vs. =90)	2.12 (1.14 to 3.93)	**0.017**
eGFR (=30 and <60 vs. =90)	2.41 (1.21 to 4.83)	**0.013**
eGFR (<30 vs. =90)	3.80 (1.64 to 8.81)	**0.002**

Data is represented as odds ratio (95. Confidence interval); *p*-value < 0.05 was considered significant. Not significant data not shown; eGRF = estimated glomerular filtration rate (calculated).

**Table 4 jcm-11-04606-t004:** Odds ratio to receive CD-TLR for all risk factors in the three sensitivity analyses.

Risk Factor	Multiple Imputation (180 Days)		Available Case		Complete Case	
	Odds Ratio (95% CI)	*p*-Value	Odds Ratio (95% CI)	*p*-Value	Odds Ratio (95% CI)	*p*-Value
Female gender	1.75 (1.15 to 2.66)	0.009	1.69 (1.11 to 2.57)	0.014	0.18 (0.01 to 4.27)	0.30
Age (per decade)	0.67 (0.54 to 0.84)	<0.001	0.67 (0.54 to 0.83)	<0.001	0.59 (0.43 to 0.83)	**0.002**
Dyslipidemia	1.82 (1.03 to 3.23)	0.041	1.89 (1.08 to 3.52)	0.034	2.11 (1.05 to 4.76)	**0.05**
eGFR (=60 and <90 vs. =90)	2.14 (1.16 to 3.97)	0.016	2.18 (1.19 to 4.11)	0.014	2.14 (1.08 to 4.33)	**0.031**
eGFR (=30 and <60 vs. =90)	2.36 (1.18 to 4.71)	0.015	2.40 (1.21 to 4.87)	0.013	2.50 (1.15 to 5.52)	**0.022**
eGFR (<30 vs. =90)	3.72 (1.61 to 8.63)	0.002	3.86 (1.64 to 8.93)	0.002	5.55 (1.85 to 16.19)	**0.002**
Female gender × age (per decade)					1.38 (0.90 to 2.15)	0.14

## Data Availability

Data is contained within the article or Appendix A.

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
