# Peer review of "Preoperative Factors Associated with Target Lesion Revascularization following Endovascular Therapy of the Superficial Femoral Artery"

_jcm, 2022, doi:10.3390/jcm11154606_

Round 1

Reviewer 1 Report

A retrospective study in which the authors try to determine the risk factors for re-revascularization in 2 groups of patients.

A / 1st group of patients is the group without TLR - n = 662 patients

B / 2nd group of patients with at least 1 TLR - n = 119.

The authors conducted an interesting statistical analysis using numerous epidemiological, numerous and laboratory data.

Remarks:

no information on group homogeneity - no information on:

a / the degree of advancement of the lesions in the treated arteries (lesion length, occlusion length or % stenosis);

b / length of implanted stents, % drug balloons (DEB) or drug stents (DES) used;

c /% of other techniques used - atherectomy, lithotripsy and others before using DEB/DES/stents.

Unfortunately, the frequency of reintervention in SFA is high and in this case it may be related not only to the factors mentioned by the authors, but also to the procedure used.

Author Response

Response to Reviewer #1

A retrospective study in which the authors try to determine the risk factors for re-revascularization in 2 groups of patients.

A / 1st group of patients is the group without TLR - n = 662 patients

B / 2nd group of patients with at least 1 TLR - n = 119.

The authors conducted an interesting statistical analysis using numerous epidemiological, numerous and laboratory data.

Remarks:

no information on group homogeneity - no information on:

a / the degree of advancement of the lesions in the treated arteries (lesion length, occlusion length or % stenosis);

b / length of implanted stents, % drug balloons (DEB) or drug stents (DES) used;

c /% of other techniques used - atherectomy, lithotripsy and others before using DEB/DES/stents.

Unfortunately, the frequency of reintervention in SFA is high and in this case it may be related not only to the factors mentioned by the authors, but also to the procedure used.

Response:

We would like to thank the reviewer for his remarks. The two groups (TLR and nonTLR) were quite homogenous regarding traditional cardiovascular risk factors, lab results or frequency of cardiovascular or cerebrovascular disease. Only for female gender, age or dyslipidema we found a difference between both groups, thus included these in the primary and secondary analyses. The information on both groups are listed in Table 2 and we further added a paragraph in the results section (marked in red i.e. track changes) of the manuscript.

However, information on lesion length, occlusion length or % stenosis or on length of implanted stents, % drug balloons (DEB) or drug stents (DES) used were not primarily searched for in our database, thus not included in the study or analyses.

The aim of the study was to detect a phenotype of risk factors in patients with symptomatic PAOD and endovascular treatment of the SFA that might predispose to mid-term TLR within 1 year after primary intervention, since non-invasive risk prediction models for TLR in PAOD are not addressed in current guidelines. Due to the retrospective character of the study, patients could not be matched regarding their interventions, unless by a propensity score. Matching is an advantage in randomized controlled trials for better comparison of inhomogenous populations. Thus, we deliberately avoided information on technical aspects of the interventions to enter our analyses. The lack of this information is truly a limitation, which we believe to have sufficiently addressed in the limitation section of the discussion. Truly, the frequency of reintervention in the TLR is high and we are well aware of the fact that the reason for TLR is only in part explained by our results.  

From the 662 patients without TLR 403 patients received a stent (61%). In contrast, 98 of the 119 patients with at least one reintervention received a stent during the index procedure (82%). This is clearly a significant difference potentially explaining a higher rate of reintervention in this group. However, in a recent publication from our department, Haine et al., described comparable results of Zilver-PTX stents vs Supera stents regarding CD-TLR in femoropoliteal lesions, not allowing to favor one stent over the other, not even for calcified or popliteal artery lesions [1].   

a / the degree of advancement of the lesions in the treated arteries (lesion length, occlusion length or % stenosis);

b / length of implanted stents, % drug balloons (DEB) or drug stents (DES) used;

a / and b /: any more detailed information on lesion length, stents or DEB/DES is of course collected in the database of the Clinic for Angiology at University Bern and can be retrieved if the reviewer sees an absolute necessity to do so. The databases setup and focus in Bern, however, is mainly on risk factors. Therefore, the search for the requested information can be done by a specific inquiry. Unfortunately, this requires time. Nevertheless, the aim of the study was to focus on identifying predisposing risk factors for TLR and not on the technical aspects.  

c /% of other techniques used - atherectomy, lithotripsy and others before using DEB/DES/stents.

c /: any techniques to debulk lesions or occlusions prior to DEB/DES or stents, like artherectomy or lithotripsy, were not used in the cases we included in the study. These techniques are not commonly used in the University hospital Bern (<1% of cases in general) due to the more conservative interventional approach/ philosophy in this department, unlike in German high-volume centers e.g. Bad Krozingen or University of Leipzig. However, rotational thrombectomy (e.g. RotarexÒ thrombectomy catheters) was often used, especially in re-interventions (stent occlusion). To retrieve information regarding rotational thrombectomy on the frequency from the database at our institution however would need time(see comments above).

Reference

[1] Haine A, Schmid MJ, Schindewolf M,  Lenz A, Bernhard SM, Drexel H, Baumgartner I, Dopheide JF. Comparison Between Interwoven Nitinol and Drug Eluting Stents for Endovascular Treatment of Femoropopliteal Artery Disease. Eur J Vasc Endovasc Surg. 2019 Dec;58(6):865-873. doi: 10.1016/j.ejvs.2019.09.002.

Reviewer 2 Report

Dear author, 

thank you for the opportunity to review this paper. I really agree that there are many preoperative cardiovascular risks that contribute to TLR and need adequately to be treated. However, my main concern for this paper is related to the type of the endovascular approach used for the recanalization of the superficial femoral artery (SFA) and is not reported. All analyses, apparently have been gathered for the patients who underwent endovascular recanalization of the SFA without considering the technique and the devices used and this is not correct. 

It is well known that a simple angioplasty with a plain ballon at one-year freedom from TLR is poorer than those performed with a drug-coated balloon. Moreover, the use of a stent, even a bare-metal stent, has improved the results in the SFA reaching an 80% of primary patency at the femoropopliteal segment. The use of the PTX eluted stents ( Zilver PTX and Eluvia) have further improved the outcomes. 

Finally, no information is reported in regard to the type of the lesion according to the TASC classification. TASC A and B have completely different outcomes from lesions TASC C and D. 

Without this information, it is difficult to perform any review. 

Kind regards. 

Author Response

Reviewer's Comments:

Dear author, 

thank you for the opportunity to review this paper. I really agree that there are many preoperative cardiovascular risks that contribute to TLR and need adequately to be treated. However, my main concern for this paper is related to the type of the endovascular approach used for the recanalization of the superficial femoral artery (SFA) and is not reported. All analyses, apparently have been gathered for the patients who underwent endovascular recanalization of the SFA without considering the technique and the devices used and this is not correct. 

It is well known that a simple angioplasty with a plain ballon at one-year freedom from TLR is poorer than those performed with a drug-coated balloon. Moreover, the use of a stent, even a bare-metal stent, has improved the results in the SFA reaching an 80% of primary patency at the femoropopliteal segment. The use of the PTX eluted stents (Zilver PTX and Eluvia) have further improved the outcomes. 

Finally, no information is reported in regard to the type of the lesion according to the TASC classification. TASC A and B have completely different outcomes from lesions TASC C and D. 

Without this information, it is difficult to perform any review. 

Kind regards. 

Response:

We like to thank the reviewer for his comments on our study. We agree with his/ her perception of SFA interventions, especially in regard to PTx coated / eluting devices.

The aim of our study was to identify a phenotype of risk factors in patients with symptomatic PAOD and endovascular treatment of the SFA that might predispose to mid-term TLR within 1 year after primary intervention, since non-invasive risk prediction models for TLR in PAOD are not addressed in current guidelines. These patients then could be informed specifically before their first intervention of their potentially higher risk for a reintervention.

The database setup of the clinic of angiology at University Bern focusses on risk factors. However, information on lesion character, angioplasty with or without a DCB or stent used (BMS or DES) has been stored, but requires a specific, time-consuming inquiry. Basic information whether a stent was used or not was simple to retrieve. From the patients with no TLR 403 patients received a stent (61%). In contrast, 98 of the 119 patients with at least one reintervention received a stent during the index procedure (82%). We have added the information to Table 2 in the manuscript (marked in red i.e. track changes). This is clearly a significant difference but does not exclusively explain the higher rate of reinterventions in this group. In a previous study from our department, Haine et al., described comparable results of Zilver-PTX stents vs Supera stents regarding CD-TLR in femoro-politeal lesions, not allowing to favor one stent, not even for calcified or popliteal artery lesions [1].

Admittingly, inclusion of our patients after an intervention creates a potential selection bias as patients with severely advanced disease and poor overall prognosis may have been excluded from intervention. Confounding changes in technical aspects during the relatively long inclusion period as well as those patients not being followed in our study center introduce another selection bias. Furthermore, our focus in the present study does not address morphological or anatomical aspects of the culprit lesion. It is unknown in which way these features might confound the study results.

Thus, the advantage of our retrospective “all inclusive” analysis allows to examine the outcome for the influence of many clinical variables under “real-life” conditions of clinical practice, whereas inclusion of further technical aspects, with the necessity to match by a propensity score, might have prevented the identification of age, female gender and renal insufficiency as predisposing risk factors of TLR.

We hope that our “preinterventional” focus, though retrospectively, is genuinely comprehensible.

Reference:

[1] Haine A, Schmid MJ, Schindewolf M,  Lenz A, Bernhard SM, Drexel H, Baumgartner I, Dopheide JF. Comparison Between Interwoven Nitinol and Drug Eluting Stents for Endovascular Treatment of Femoropopliteal Artery Disease. Eur J Vasc Endovasc Surg. 2019 Dec;58(6):865-873. doi: 10.1016/j.ejvs.2019.09.002.

Round 2

Reviewer 2 Report

I read the revised version and I was not surprised to see that patients having stents had better clinical-driven TLR. However, I have a lot of criticisms for this paper. 

Major aspects:

1.     The authors did not respond to my question and did not make any change for explaining the question I raised concerning SFA stenting,

2.     No mention in regard to the TASC II lesion, but the authors cited Tosaka et al in their discussion reporting stratification of the patients in three subgroups in regard to the type of lesion. Moreover, they include in the discussion the Tosaka’s conclusion “Except for cerebrovascular disease, the authors found no statistical significance in baseline characteristics among classes and  concluded, that re-stenotic patterns and reference vessel diameters after femoropopliteal stenting are independent predictors of recurrent TLR and occlusion” These characteristics are still missing in the Methods section

3.     Author cited trials such as Japanese Zilver PTX and Zeller who reported data on patients treated with drug-eluting stents comparing two clinical arms, but in this study, all procedures are put together. For another example, the authors cited Suzuki. Sukuki’s paper is evaluated the outcomes in patients treated all with one specific stent, namely the Control TM stent

Minor aspects:

1.     Authors state "Decision for TLR was clinically driven" which means that they are reporting data on clinical driven TLR. I would suggest modifying TLR with cdTLR

2.     In the conclusion I would change the term “tremendously” with another adverb (extremely, substantially).

3.     In the conclusion the statement “Keeping these limitations in mind, we were able to show that between 1 and 12 months after the intervention of the SFA subjects of younger age, of female gender and with dyslipidemia have a higher odds ratio of receiving multiple interventions than male subjects or subjects without dyslipidemia and that the interaction term between age and gender hinting towards a lower risk for multiple interventions in young female” is very redundant. The meaning is merely that young old women and patients with dyslipidemia are at risk for cdTLR.

4.     The statement in the Discussion “but in contrast hypercholesterolemia was

not found predictive [18], which might be explained in part by the higher prevalence of dyslipidemia in our cohort (in total: 66.5% vs 77.5%).” this is in contrast to the authors’ conclusion.

5.     The discussion is too long 

Author Response

Kind regards
